# Effects of Proton Pump Inhibitors on Patient Survival in Patients Undergoing Maintenance Hemodialysis

**DOI:** 10.3390/jcm12144749

**Published:** 2023-07-18

**Authors:** Seok Hui Kang, Gui Ok Kim, Bo Yeon Kim, Eun Jung Son, Jun Young Do

**Affiliations:** 1Division of Nephrology, Department of Internal Medicine, College of Medicine, Yeungnam University, Daegu 42415, Republic of Korea; kangkang@ynu.ac.kr; 2Healthcare Review and Assessment Committee, Health Insurance Review and Assessment Service, Wonju 26465, Republic of Korea

**Keywords:** dialysis, hemodialysis, mortality, proton pump inhibitor

## Abstract

Data to draw definite conclusions regarding the association between proton pump inhibitor (PPI) and all-cause mortality in patients undergoing hemodialysis (HD) remain insufficient. The object of this retrospective study was to assess the impact of PPIs on patient survival within a substantial cohort of individuals receiving maintenance HD. To achieve this, the study employed laboratory and clinical data sourced from the 4th, 5th, and 6th National HD Quality Assessment Programs. The programs included patients undergoing maintenance HD (n = 54,903). Based on the PPI prescription data collected over the 6-month HD quality assessment, the patients were categorized into three groups: Group 1, comprising individuals with not prescription; Group 2, consisting of patients prescribed PPIs for less than 90 days; and Group 3, comprising patients prescribed PPIs for 90 days or more. The respective number of patients in Groups 1, 2, and 3 was 43,059 (78.4%), 5065 (9.2%), and 6779 (12.3%), respectively. Among the study groups, the 5-year survival rates were as follows: Group 1—70.0%, Group 2—68.4%, and Group 3—63.0%. The hazard ratio for Group 3 was 1.09 (95% CI, 1.04 to 1.15; *p* < 0.001) and 1.10 (95% CI, 1.03 to 1.18; *p* = 0.007) compared to Groups 1 or 2 based on multivariable analysis. Multivariable analyses revealed a lower rate of patient survival in Group 3 compared to the other groups, while Groups 1 and 2 exhibited similar patient survival rates. Our study revealed a significant association between long-term PPI usage and increased mortality among patients undergoing HD. However, distinct trends were observed in subgroup analyses. The association between long-term PPI usage and mortality was prominent in patients who did not have a high gastrointestinal burden or comorbidities. Meanwhile, this association was not observed in patients who did have a high gastrointestinal burden or comorbidities.

## 1. Introduction

Hemodialysis (HD) remains the predominant modality utilized for patients with end-stage renal disease necessitating renal replacement therapy. Recent registry data have shown that the proportion of HD among the three modalities were approximately 67% in the USA and 81% in South Korea [1,2]. High mortality in patients undergoing HD compared to that in the general population not undergoing HD is well known [3]. The risk factors for mortality in patients undergoing HD include anemia, chronic kidney disease-mineral and bone disease, chronic inflammation, malnutrition, vascular calcification, and other comorbidities. Several well-established factors associated with mortality in patients undergoing HD have been identified in prior studies. However, considering the high mortality rate of patients undergoing HD, research to identify new risk factors is ongoing.

Patients receiving HD commonly experience polypharmacy; a previous study revealed that patients undergoing HD on average take 12 pills per day [4]. Proton pump inhibitors (PPIs) are the most common type of medication used by patients undergoing HD [5]. The use of PPIs by patients undergoing HD is approximately 57% in Japan and 63% in the USA [6,7]. Patients undergoing HD exhibited a higher prevalence of gastrointestinal problems compared to the general population, and PPIs may be useful in treating various gastrointestinal diseases such as dyspepsia, peptic ulcer, and gastroesophageal reflux disease. However, previous studies have shown that the use of PPIs are associated with an increased high risk of hypomagnesemia, osteoporosis, hip fractures, and aortic calcification in patients undergoing HD [8,9,10,11,12,13]. Other studies have reported a positive association between PPI usage and all-cause mortality among patients undergoing HD [6,13,14]. However, the available data regarding the association between PPI usage and all-cause mortality in patients undergoing HD are currently inadequate to establish definitive conclusions. In addition, regional and ethnic differences regarding the association between two variables should also be considered. This study aimed to evaluate the effect of PPIs on patient survival in a large sample of patients undergoing maintenance HD.

## 2. Materials and Methods

### 2.1. Data Source and Population

The data for this retrospective study was gathered from both the National HD Quality Assessment Program (4th, 5th, and 6th), encompassing laboratory and clinical data, and the Health Insurance Review and Assessment (HIRA), providing claims data [15,16]. The inclusion criteria for these programs consisted of patients who underwent HD for a minimum of three months, received HD at least twice a week (eight times per month), and were 18 years of age or older. The study followed ethical guidelines outlined in the World Medical Association’s Declaration of Helsinki and received approval from the institutional review board of Yeungnam University Hospital (approval no. YUMC 2022-01-010). As this was a retrospective study with anonymized and de-identified participant records, obtaining informed consent was not required in this study.

In the 4th, 5th, and 6th HD Quality Assessment Programs, 21,846, 35,538, and 31,294 patients were included, respectively (Appendix A). Participants who had participated in the study previously (*n* = 32,440), those with incomplete datasets, and those who underwent HD utilizing a dual lumen catheter (*n* = 1335) were excluded from the analysis. This resulted in a final inclusion of 54,903 patients for this study. Patients with an HD catheter may have reversible renal diseases and/or other acute complications. Furthermore, HD catheters themselves are also associated with various complications, which can be confounding factors for accurately determining all-cause mortality. Therefore, we have excluded all patients with HD catheters in our cohort. The remaining participants exclusively received HD through an arteriovenous fistula or graft. Significantly, over the 6-month HD quality assessment period, it is worth mentioning that none of the patients required blood transfusions, indicating the absence of any instances of active bleeding. 

### 2.2. Study’s Variables

The clinical data gathered included information concerning age, the underlying cause of end-stage renal disease, gender, and type of vascular access. Additionally, laboratory data obtained during the assessment included hemoglobin levels (g/dL), Kt/V_urea_ values, serum albumin concentrations (g/dL), serum calcium levels (mg/dL), serum phosphorus levels (mg/dL), serum creatinine (SCr) levels (mg/dL), predialysis systolic blood pressure (SBP, mmHg), predialysis diastolic blood pressure (DBP, mmHg), and ultrafiltration volume per session (UFV, L). These data were collected on a monthly basis and subsequently averaged. We calculated Kt/V_urea_ using Daugirdas’ equation [17].

Based on the PPI prescription data collected over the six-month HD quality assessment, the patients were categorized into 3 groups. The medication codes used are listed in Appendix A. Group 1 included patients with no prescriptions during the assessment period. Group 2 comprised patients who were prescribed PPIs for less than 90 days during the assessment period. Group 3 encompassed patients who were prescribed PPIs for 90 days or more during the assessment period. Furthermore, the concurrent usage of antihypertensive drugs, aspirin, and statins was assessed, with their use defined as the presence of one or more prescriptions during the six months of each HD Quality Assessment Program.

Comorbidity assessment occurred one year prior to the HD Quality Assessment Program, employed the comorbidity codes established by Quan et al. [18,19]. In our study, the Charlson Comorbidity Index (CCI), which encompasses 17 comorbidities, was applied. As all patients were undergoing HD due to renal disease, they were automatically considered to have renal disease as a comorbidity. After comorbidities were defined and identified, CCI scores were calculated. We used coronary artery disease (CAD), gastrointestinal disease (GID), steroid usage, and antiplatelet agents to define subgroups for analysis. We defined CAD as the presence of a procedural code for percutaneous transluminal coronary angioplasty within 1 year prior to the HD Quality Assessment Program (M6551, M6552, M6561-4, M6571, and M6572). We defined GID using the procedural code for the upper gastrointestinal tract rather than the ICD-10 disease code, due to inaccuracies in its ability to diagnose GID. GID was determined by the existence of procedural codes indicating upper gastrointestinal perforation endoscopic treatment (Q7660), simple closure of perforated stomach and duodenum (Q2540), surgical clipping (Q2510), endoscopic hemostasis of the upper gastrointestinal tract (Q7620), and embolization (M6644). Steroid usage was defined as an oral steroid prescription for ≥30 days during the assessment period. Antiplatelet agent usage was defined as a prescription for aspirin, clopidogrel, or ticlopidine for ≥30 days during the assessment period.

Patient follow-up was conducted until April 2022; if a patient underwent peritoneal dialysis (PD) or kidney transplantation (KT), the date of that event marked the endpoint of their follow-up period, with the data collected after that date being censored. The study used electronic data to track the clinical outcomes of patients, excluding death. Information regarding patient mortality was acquired from the HIRA. Specific codes, such as O7072, O7071, and O7061 for PD and R3280 for KT, were used to indicate that a patient was censored from the study, meaning they were no longer included in the study.

### 2.3. Statistical Analyses

The data was analyzed using two statistical software programs: SAS Enterprise Guide (version 7.1) and R (version 3.5.1). Categorical variables were summarized as counts and proportions, while continuous variables were summarized as means and standard deviations. Categorical variables were compared to see if there was a significant association between them. Pearson’s chi-square test or Fisher’s exact test was used, depending on the size of the dataset. For continuous variables, the mean was compared between groups using a one-way analysis of variance. The Tukey post-hoc test was then used to identify which groups were significantly different. Survival estimates were derived from two statistical methods: Cox regression analyses and Kaplan–Meier curves. Utilizing the log-rank test, *p*-values were computed to assess the disparities between survival curves.

Utilizing Cox regression analysis, the hazard ratio (HR) and its corresponding 95% confidence interval (CI) were determined. Adjustment for confounders was performed in the multivariable Cox regression analyses including age, the underlying cause of end-stage renal disease, sex, vascular access type, CCI score, HD vintage, UFV, SBP, DBP, Kt/V_urea_, hemoglobin, serum albumin, SCr, serum calcium, serum phosphorus, and use of antihypertensive drugs, aspirin, or statins, and were performed using the enter mode. We selected all variables associated with patients’ survival as covariates. All baseline characteristics included in the analysis are widely recognized as factors associated with patient survival and were chosen as covariates.

Most baseline characteristics differed between the 3 groups. Propensity score weights were utilized for our analyses. To obtain these weights for the three groups, we employed generalized boosted models, incorporating the following variables: vascular access type, underlying cause of end-stage renal disease, age, CCI score, gender, HD vintage, UFV, SBP, DBP, Kt/V_urea_, hemoglobin, serum albumin, SCr, serum calcium, serum phosphorus, and the use of statins, aspirin, or antihypertensive drugs. We deemed results to be statistically significant if the corresponding *p* value were less than 0.05.

## 3. Results

### 3.1. Participants

The number of patients in Groups 1, 2, and 3 was 43,059 (78.4%), 5065 (9.2%), and 6779 (12.3%), respectively. Group 3 had a higher mean age than the other groups (Table 1).

In comparison to that of other groups, Group 1 exhibited a longer follow-up duration, higher UFV, and higher hemoglobin, serum albumin, serum phosphorus, serum calcium, and SCr levels. Additionally, Group 1 had a greater proportion of patients with arteriovenous fistula and a larger number of male participants than that in other groups. Meanwhile, Group 3 had a higher CCI score, a higher proportion of patients with diabetes mellitus (DM), and greater utilization of antihypertensive drugs, aspirin, and statins compared to that of other groups.

### 3.2. Survival

At the end of the follow-up period, the number of patients in the survivor, death, PD, and KT subgroups was 22,988 (53.4%), 16,504 (38.3%), 144 (0.3%), and 3423 (7.9%) in Group 1; 2680 (52.9%), 1990 (39.3%), 14 (0.3%), and 381 (7.5%) in Group 2; 3459 (51.0%), 2939 (43.4%), 31 (0.5%), and 350 (5.2%) in Group 3, respectively (*p* < 0.001). Group 3 displayed a heightened occurrence of deaths at the follow-up endpoint in relation to the other groups. The 5-year survival rates of Groups 1, 2, and 3 were 70.0%, 68.4%, and 63.0%, respectively (Figure 1; *p* < 0.001 for trend; *p* = 0.003 for Group 1 vs. 2; *p* < 0.001 for Group 3 vs. Groups 1 or 2).

Cox regression analyses showed that the HR for Group 3 was 1.28 (95% CI, 1.23 to 1.33; *p* < 0.001) and 1.19 (95% CI, 1.12 to 1.26; *p* < 0.001) compared to Groups 1 or 2 based on univariate analysis (Table 2).

The HR for Group 3 was 1.09 (95% CI, 1.04 to 1.15; *p* < 0.001) and 1.10 (95% CI, 1.03 to 1.18; *p* = 0.007) compared to Groups 1 or 2 based on multivariable analysis. Multivariable analyses revealed a lower rate of patient survival in Group 3 compared to the other groups, while Groups 1 and 2 exhibited similar patient survival rates. Additionally, subgroup analyses were conducted based on age, sex, and the presence of DM, as depicted in Figure 2.

Although overall trends in subgroups resembled those in the entire cohort, statistical significance was higher in female patients, young patients, or patients without DM. We additionally performed subgroup analyses based on the presence of CAD or GID, as well as steroid and antiplatelet agent usage (Appendix A). Multivariable analyses showed that PPI usage was not associated with patient survival in patients with CAD or GID, or who used steroid or antiplatelet agent. However, Group 3 showed poorer survival rates than that of other groups among patients without CAD or GID, and in those who did not use steroids or antiplatelet agents.

### 3.3. Analyses That Used Propensity Score Weights

The clinical characteristics of the three groups were compared, and Kaplan–Meier survival rates were evaluated using appropriate sampling weights. To assess the balance among the 3 groups, we calculated the maximum pairwise absolute standardized mean differences (AMDs) of the covariates before and after weighting (refer to Appendix A). Most covariates exhibited decreased maximum AMDs and differences in clinical characteristics after weighting (refer to Appendix A). The Kaplan–Meier curves demonstrated 5-year survival rates of 69.2% for Group 1, 69.6% for Group 2, and 67.2% for Group 3 (Appendix A). Group 1 and Group 2 demonstrated superior survival rates compared to Group 3 (*p* < 0.001 for trend). However, the differences in survival rates between the groups were smaller when the data were weighted than when they were not weighted.

Weighted Cox regression analyses showed that, on univariate analyses, Group 2 had a hazard ratio of 0.99 (95% CI, 0.97 to 1.01, *p* = 0.287), and Group 3 had a hazard ratio of 1.06 (95% CI, 1.04 to 1.08, *p* < 0.001), both compared to Group 1. The hazard ratio for Group 3, compared to Group 2, was 1.07 (95% CI, 1.05 to 1.09, *p* < 0.001). Among the three groups, Group 1 exhibited the lowest patient survival, while the survival rates for Groups 1 and 2 were not significantly different. Following adjustment for confounding factors, the hazard ratios for death were 1.00 (95% CI, 0.98 to 1.03, *p* = 0.814) in Group 2 and 1.08 (95% CI, 1.06 to 1.11, *p* < 0.001) in Group 3, both compared to Group 1. The hazard ratio for death was 1.08 (95% CI, 1.05 to 1.10, *p* < 0.001) in Group 3 compared to Group 2. Multivariable Cox regression analyses demonstrated consistent trends in the univariate analyses. Moreover, the results of the weighted data analyses closely resembled those obtained from the entire cohort data.

## 4. Discussion

Our study included 54,903 patients undergoing maintenance HD. The findings revealed that 21.5% of patients undergoing HD used PPIs, and with 12.3% of them using PPIs in the long term. Group 3 with long-term PPI use had higher mortality rates than Groups 1 or 2 based on Kaplan–Meier analyses. These results were similar to those obtained from multivariable Cox regression analyses. In the subgroup analyses, the statistical significance differed in each subgroup, despite similar patterns observed to those of the total cohort. Furthermore, we conducted analyses using propensity score weighting, which yielded results comparable to those obtained from the unweighted data.

Previous studies have shown a positive relationship between PPI use and mortality in patients undergoing HD. Kosedo et al. analyzed data from a Japanese registry of 376 patients undergoing HD (217 with and 159 without PPIs) [6]. They analyzed the association between PPI use and composite outcomes, including all-cause mortality, nonfatal myocardial infarction, and nonfatal stroke. Their results showed that patients who used PPIs had higher composite outcome rates than those who did not use PPIs. Francisco et al. evaluated data from a multicenter cohort of 2442 patients undergoing HD [14]. In their study, 79.2% of the total cohort used PPIs. They showed that patients using PPIs had HR 1.37 (*p* = 0.02) compared to those who did not use PPIs, based on multivariable Cox regression analysis. A prospective single-center study was performed in Japan, whereby they evaluated 399 patients undergoing HD. The study demonstrated an inverse correlation between PPI usage and serum magnesium levels [13]. Furthermore, the use of PPIs was linked to all-cause mortality as a result of diminished magnesium levels. These studies have consistently shown a positive association between PPI usage and all-cause mortality in patients undergoing HD. However, in these studies, PPI use was defined at the baseline, and the sample size and follow-up duration were limited, which may have affected the generalizability of the findings.

Several notable strengths are present in our study, including the incorporation of a sizable sample encompassing 46% of the total population of patients undergoing HD in South Korea during the corresponding HD quality assessment [2]. In addition, our study included the laboratory data associated with HD adequacy despite a large sample size and claims data. The use of PPIs was contemplated based on prescriptions during each HD quality assessment period, and patients with prescriptions of 90 days were considered under Group 3 with long-term prescriptions. With these advantages, our study can help overcome the constraints of previous investigations and establish a more definitive connection between the utilization of PPIs and patient death.

This study also provides information on regional and ethnic issues. First, the proportion of PPI use in South Korea was relatively low compared to that in other countries (57% in Japan and 63% in the USA). The low PPI prescription rate may be associated with the case payment system in the country, under which the cost of HD is paid according to the case rather than for performance. The case payment system is a payment model in which a fixed cost is paid per HD session, covering all interventions including medications, laboratory studies, and dialyzers. In the case payment system, fewer prescriptions of medications are associated with better profits. Therefore, clinicians may avoid prescribing unnecessary medications to patients unless they are essential. This finding may be associated with the low prescription of PPIs in Republic of Korea. Second, PPI usage was associated with high mortality in our study, and the risk was higher in patients with long-term use alone. In our study, Group 3 had a higher mortality than the other groups, but there was no significant difference in patient mortality between Group 1 and Group 2. These findings reveal that discontinuing PPIs at the appropriate time does not significantly affect survival rates, and the hazardous effects of long-term use of PPIs can be avoided. Third, the effect of PPIs on patient survival rates was more pronounced in low-risk patients, including female patients, young patients, and patients without DM. These findings revealed that the benefits of PPIs were greater for high-risk patients than for low-risk patients. The benefits for high-risk patients may attenuate the hazardous effects of PPIs. However, the trend of high mortality associated with the use of PPIs remained consistent across all groups of patients undergoing HD, indicating that the non-essential PPI usage should be avoided in this population.

Results of subgroup analyses based on the presence of CAD or GID, or steroid or antiplatelet agent usage, indicate that PPI usage is not consistently associated with higher mortality in patients undergoing HD. We were unable to confirm whether the presence of CAD or GID, or steroid or antiplatelet agent usage are absolute indications of PPI usage. However, steroid or antiplatelet agent usage, or the presence of CAD or GID may partially explain the reason for PPI usage. The association between PPI usage and mortality was not observed in patients with a higher burden of gastrointestinal problems, including those with CAD, GID, or who were using steroid or antiplatelet agents. This suggests that PPI use may be acceptable only in these specific patient subgroups.

The benefits or risks of using PPIs in patients undergoing HD with comorbidities or medications that increase the risk of gastrointestinal (GI) bleeding remain inconclusive. Patients with CAD often take mono or dual antiplatelet agents, but not all of them require PPIs. PPIs can potentially impact the pharmacokinetics of antiplatelet agents, leading to an increased risk of cardiovascular events or mortality. Previous studies have indicated that the concurrent use of PPIs and antiplatelet agents is associated with unfavorable outcomes despite a decrease in GI bleeding [20]. Consequently, the use of PPIs in patients with CAD or those taking antiplatelet agents is typically recommended for high-risk individuals who solely face the risk of GI bleeding. In our study, among patients with CAD or those taking antiplatelet agents, the use of PPIs did not show a significant difference in patient survival compared to those who did not use PPIs. The observation could be attributed to the inherent high risk of GI bleeding in patients undergoing HD [21]. In patients with CAD or those taking antiplatelet agents, the potential benefits of PPIs may mitigate their potential risks, resulting in a lack of association between PPI use and patient survival. Conversely, among patients without CAD or antiplatelet agents, the use of PPIs was associated with poorer patient survival compared to those without PPI use. In patients without CAD or antiplatelet agents, the potential risks of PPIs risks may outweigh their potential benefits. These associations may also apply to patients with underlying GID or those receiving steroid therapy. Recent guidelines have indicated that steroid therapy does not significantly harm the GI mucosa, and the concomitant use of steroids and PPIs is recommended for patients with a history of peptic ulcer or non-steroidal anti-inflammatory drug use [22]. In our study, among patients undergoing HD with GID or receiving steroid therapy, such as CAD or antiplatelet agents, the benefits of PPIs may attenuate their potential risks, resulting in a lack of association between PPI use and patient survival. Most guidelines or studies concerning the use of PPIs have been based on results obtained from the general population. Further research is necessary to determine whether the use of PPIs is beneficial or hazardous in patients undergoing HD with comorbidities or medications associated with an increased risk of bleeding.

In our study, the mean ages of Groups 1, 2, and 3 were 59.8, 60.2, and 62.6 years old, respectively. These ages are approximately 2–3 years younger than those from Japanese data and other Korean registries [2,23]. The National HD Quality Assessment Program included patients who underwent HD in the outpatient department, excluding those admitted for acute events that could influence HD quality. Elderly patients undergoing HD have a high prevalence of physical disability. Some of these elderly patients underwent HD on admission to convalescent hospitals; such patients would be expected to have a higher prevalence of admission due to acute/chronic events during the duration of each assessment compared to that of younger patients. Thus, the exclusion of admitted patients may have contributed to the relatively younger age observed in our cohort. In addition, DM was the most common underlying condition that led to end-stage renal disease in our cohort. In the Korean registry, the frequency of DM as the cause of end-stage renal disease was 19.5% in 1992, 30.5% in 2005, and 47.0% in 2021, respectively [2]. The prevalence of DM is steeply increasing. Considering that 9–11% of our cohort have an unknown cause for end-stage renal disease, approximately half of HD patients are undergoing HD due to DM.

In our study, mean hemoglobin levels ranged from 10.5–10.6 g/dL. Previous guidelines recommended target hemoglobin levels of 10–11.5 g/dL, with levels above 13 g/dL not being recommended [24]. In our cohort, the number of patients with hemoglobin levels <10 g/dL was 7585 (13.8%). The relatively high proportion of patients with hemoglobin levels below the target range may be due to difficulty of obtaining erythropoietin-stimulating agents. In South Korea, insurance covers erythropoiesis-stimulating agents in patients undergoing HD with hemoglobin levels less than or equal to 11 g/dL. If the patient undergoing HD has hemoglobin levels greater than 11 g/dL, the clinician may discontinue the erythropoiesis-stimulating agent, even though the hemoglobin levels are expected to decrease. Therefore, maintaining a stable hemoglobin level >10–11 g/dL is challenging due to insurance limitations. Nevertheless, the mean hemoglobin levels in all three groups of our cohort remained within the target range.

Our study has some drawbacks. One limitation of our study is that it was retrospective. Second, comorbidities and medication usage relied solely on claims data, which may introduce discrepancies between prescribed medications and their actual utilization. Third, we were unable to assess the association between PPI use and gastrointestinal diseases (e.g., gastritis, gastroesophageal reflux disease, or peptic ulcers). In South Korea, insurance coverage for PPIs includes treatment/prophylaxis of gastric/duodenal ulcers, eradication of *Helicobacter pylori*, and treatment/maintenance of gastroesophageal reflux disease, and the ICD-10 codes for these diseases should be entered for the prescription of PPIs. Therefore, the ICD-10 codes for these diseases may be routinely entered regardless of the presence of these diseases. Nevertheless, no patient in our cohort received a transfusion during the HD quality assessment. The PPIs for most gastrointestinal diseases are prescribed for 4–8 weeks [25]. This may reveal that there were few patients with severe gastrointestinal diseases, such as bleeding, in our cohort, and most of the Group 3 patients might have used PPIs regardless of organic lesions of the gastrointestinal tract. Fourth, our study did not incorporate data concerning the reasons behind the recorded deaths. Further investigation of the causes of death, such as cardiovascular disease or bleeding, would be helpful to evaluate the effects of PPIs beyond all-cause mortality. Fifth, sample sizes and the baseline characteristics differed among the three groups, which could be a source of selection bias. However, we attempted to overcome this limitation by adjusting several covariates for multivariable and subgroup analyses. Multivariable and subgroup analyses showed comparable trends to the univariate analysis conducted on the entire cohort.

## 5. Conclusions

Our study revealed a significant association between long-term PPI usage and increased mortality among patients undergoing HD. However, distinct trends were observed in subgroup analyses based on sex, age, the presence of DM, CAD, GID, and the use of steroids or antiplatelet agents. The association between long-term PPI usage and mortality was prominent in patients who did not have a high gastrointestinal burden or comorbidities, particularly those without CAD, GID, steroid therapy, or antiplatelet agents. Meanwhile, this association was not observed in patients who did have high gastrointestinal burden or comorbidities, including those with CAD, GID, steroid therapy, or antiplatelet agents. Given the limitations of our study, it is challenging to ascertain whether PPI use is hazardous in patients undergoing HD. Therefore, further prospective randomized studies are required to clarify this association.

## Figures and Tables

**Figure 1 jcm-12-04749-f001:**
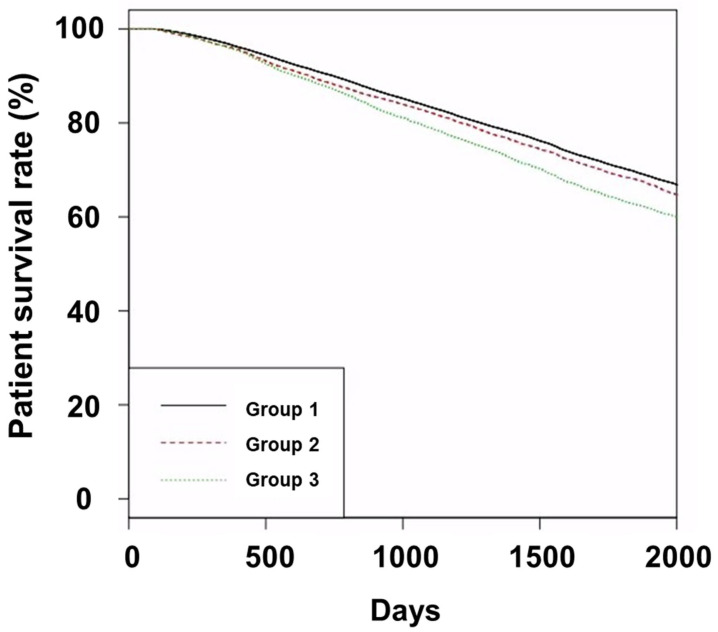
Patient survival curves.

**Figure 2 jcm-12-04749-f002:**
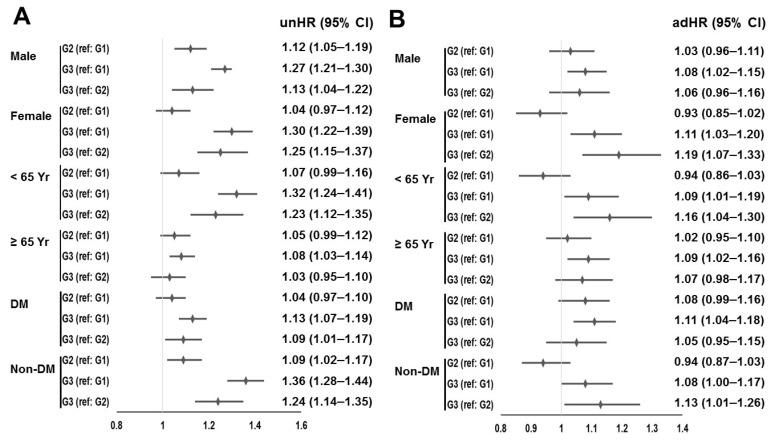
HRs and 95% CIs by subgroups, visualized in forest plots. (**A**) Univariate and (**B**) multivariable Cox regression analyses. The analyses were adjusted according to underlying cause of end-stage renal disease, vascular access type, Charlson Comorbidity Index score, age, gender, hemodialysis vintage, ultrafiltration volume, systolic blood pressure, diastolic blood pressure, Kt/V_urea_, serum albumin, hemoglobin, serum creatinine, serum calcium, serum phosphorus, the use of anti-hypertensive drugs, aspirin, and statin. Abbreviations: adHR, adjusted hazard ratio; CI, confidence interval; DM, diabetes mellitus; G1, Group 1; G2, Group 2; G3, Group 3; ref, reference; unHR, unadjusted hazard ratio; Yr, years.

**Table 1 jcm-12-04749-t001:** Clinical characteristics of patients.

	Group 1 (*n* = 43,059)	Group 2 (*n* = 5065)	Group 3 (*n* = 6779)	*p*-Value
Age (years)	59.8 ± 13.1	60.2 ± 12.6	62.6 ± 12.2 *#	<0.001
Gender (male sex, %)	26,084 (60.6%)	2768 (54.6%)	3932 (58.0%)	<0.001
Hemodialysis vintage (days)	1593 ± 1714	1562 ± 1760	1459 ± 1585 *#	<0.001
Underlying cause of ESRD				<0.001
Diabetes mellitus	18,463 (42.9%)	2257 (44.6%)	3431 (50.6%)	
Hypertension	11,510 (26.7%)	1316 (26.0%)	1583 (23.4%)	
Glomerulonephritis	4657 (10.8%)	504 (10.0%)	643 (9.5%)	
Others	3653 (8.5%)	442 (8.7%)	486 (7.2%)	
Unknown	4776 (11.1%)	546 (10.8%)	636 (9.4%)	
CCI score	7.3 ± 2.9	8.0 ± 2.8 *	8.6 ± 2.8 *#	<0.001
Follow-up duration (days)	1883 ± 874	1813 ± 849 *	1713 ± 819 *#	<0.001
Vascular access				<0.001
Autologous arteriovenous fistula	36,872 (85.6%)	4261 (84.1%)	5676 (83.7%)	
Arteriovenous graft	6187 (14.4%)	804 (15.9%)	1103 (16.3%)	
Kt/V_urea_	1.53 ± 0.27	1.53 ± 0.28	1.54 ± 0.27 *	0.002
UFV (L/session)	2.29 ± 0.96	2.24 ± 0.93 *	2.20 ± 0.94 *#	<0.001
Hemoglobin (g/dL)	10.7 ± 0.8	10.6 ± 0.8 *	10.6 ± 0.8 *	<0.001
Serum albumin (g/dL)	4.00 ± 0.34	3.95 ± 0.35 *	3.93 ± 0.35 *#	<0.001
Serum phosphorus (mg/dL)	5.0 ± 1.4	4.9 ± 1.4 *	4.6 ± 1.3 *#	<0.001
Serum calcium (mg/dL)	8.9 ± 0.8	8.9 ± 0.8 *	8.8 ± 0.8 *#	<0.001
Systolic blood pressure (mmHg)	141 ± 16	142 ± 16 *	141 ± 16	0.027
Diastolic blood pressure (mmHg)	78 ± 9	78 ± 10	77 ± 10 *#	<0.001
SCr (mg/dL)	9.6 ± 2.7	9.3 ± 2.7 *	9.1 ± 2.7 *#	<0.001
Use of antihypertensive drug	28,235 (65.6%)	3956 (78.1%)	5456 (80.5%)	<0.001
Use of aspirin	17,711 (41.1%)	2464 (48.6%)	3431 (50.6%)	<0.001
Use of statin	11,637 (27.0%)	1826 (36.1%)	2782 (41.0%)	<0.001

Continuous variables are summarized as mean and standard deviation, while categorical variables are summarized as counts and proportions. Continuous variables were analyzed using one-way analysis of variance, and the Tukey post-hoc test was employed for pairwise comparisons. Categorical variables, on the other hand, were assessed using Pearson’s χ^2^ test. Abbreviations: CCI, Charlson Comorbidity Index; ESRD, end-stage renal disease; SCr, serum creatinine; UFV, ultrafiltration volume. * *p* < 0.05 vs. Group 1, # *p* < 0.05 vs. Group 2.

**Table 2 jcm-12-04749-t002:** Cox regression analyses of factors associated with patient survival.

	Univariate		Multivariable	
	HR (95% CI)	*p*	HR (95% CI)	*p*
Group				
Ref: Group 1				
Group 2	1.07 (1.03–1.13)	0.003	0.99 (0.94–1.05)	0.846
Group 3	1.28 (1.23–1.33)	<0.001	1.09 (1.04–1.15)	<0.001
Ref: Group 2				
Group 3	1.19 (1.12–1.26)	<0.001	1.10 (1.03–1.18)	0.007
Underlying disease of ESRD (Ref: diabetes mellitus)	0.81 (0.80–0.82)	<0.001	0.90 (0.88–0.91)	<0.001
Age (1-year increase)	1.06 (1.06–1.06)	<0.001	1.06 (1.06–1.06)	<0.001
Vascular access type (ref: AVF)	1.51 (1.46–1.56)	<0.001	1.19 (1.14–1.24)	<0.001
CCI score (1-score increase)	1.14 (1.13–1.14)	<0.001	1.07 (1.06–1.08)	<0.001
Gender (Ref: male sex)	0.87 (0.84–0.89)	<0.001	0.97 (0.71–0.77)	<0.001
Hemodialysis vintage (1-day increase)	1.00 (1.00–1.00)	0.183	1.00 (1.00–1.01)	<0.001
UFV (1 kg/session increase)	0.92 (0.90–0.93)	<0.001	1.07 (1.05–1.09)	<0.001
KtV_urea_ (1-unit increase)	0.91 (0.87–0.97)	0.001	0.82 (0.76–0.88)	<0.001
Hemoglobin (1 g/dL increase)	0.86 (0.85–0.88)	<0.001	0.90 (0.88–0.92)	<0.001
Serum albumin (1 g/dL increase)	0.37 (0.36–0.39)	<0.001	0.63 (0.59–0.66)	<0.001
SCr (1 mg/dL increase)	0.87 (0.86–0.87)	<0.001	0.94 (0.93–0.94)	<0.001
Serum phosphorus (1 mg/dL increase)	0.85 (0.84–0.86)	<0.001	1.04 (1.03–1.06)	<0.001
Serum calcium (1 mg/dL increase)	0.94 (0.92–0.95)	<0.001	1.07 (1.04–1.09)	<0.001
Systolic blood pressure (1 mmHg increase)	1.01 (1.01–1.01)	<0.001	1.01 (1.00–1.01)	<0.001
Diastolic blood pressure (1 mmHg increase)	0.98 (0.98–0.99)	<0.001	1.00 (1.00–1.01)	0.018
Use of anti-hypertensive drug	1.12 (1.09–1.16)	<0.001	0.95 (0.91–0.98)	0.006
Use of aspirin	1.16 (1.13–1.19)	<0.001	0.97 (0.94–1.01)	0.106
Use of statin	1.10 (1.07–1.14)	<0.001	0.96 (0.92–0.99)	0.029

Multivariable analysis was conducted after adjusting for various factors, including underlying cause of ESRD, age, vascular access type, CCI score, gender, hemodialysis vintage, systolic blood pressure, diastolic blood pressure, UFV, Kt/V_urea_, hemoglobin level, serum albumin level, SCr level, serum calcium level, serum phosphorus level, and use of anti–hypertensive drug, aspirin, and statins. The analysis was conducted using the enter mode for multivariable Cox regression. Abbreviations: AVF, arteriovenous fistula; CCI, Charlson Comorbidity Index; CI, confidence interval; ESRD, end-stage renal disease; HR, hazard ratio; Ref, reference; SCr, serum creatinine; UFV, ultrafiltration volume.

## Data Availability

The raw data were generated at the Health Insurance Review and Assessment Service. The database can be requested from the Health Insurance Review and Assessment Service by sending a study proposal including the purpose of the study, study design, and duration of analysis through an e-mail (turtle52@hira.or.kr) or at the website (https://www.hira.or.kr, accessed on 23 May 2023). The authors cannot distribute the data without permission.

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
