# Peer review of "Effects of Proton Pump Inhibitors on Patient Survival in Patients Undergoing Maintenance Hemodialysis"

_jcm, 2023, doi:10.3390/jcm12144749_

Round 1

Reviewer 1 Report

In the manuscript, the authors analyzed the use of proton pump inhibitors (PPIs) in the population of hemodialysis patients. This is a retrospective study that included a large sample size. The main conclusion is that the use of PPIs is associated with a higher risk of all-cause mortality, especially in patients with a high burden of comorbidities, but the impact of PPI side effects in this group of patients cannot be ruled out. The main drawback is that this information is practically irrelevant from a practical point of view, as no in-depth analysis of PPI indications and causes of death has been performed. We do not know whether these drugs were used for, for example, gastritis found in gastroscopy or prophylactically due to, taking glucocorticosteroids. We also do not know whether the cause of death was infectious complications or cardiovascular disease.

Such a limitation was also emphasized by the authors, which unfortunately reduces the value of this work.

After reading the manuscript, the reader still does not know whether PPIs are friends or foes of HD patients who have a higher burden of gastrointestinal problems. Of course, I understand that with such a large sample it is difficult to collect such data, but without it it is impossible to draw any practical conclusions.

However, apart from this fundamental remark, I have one minor. I think the Table 2  needs to be corrected because the lines are shifted and the HR values seem to be misplaced.

Author Response

In the manuscript, the authors analyzed the use of proton pump inhibitors (PPIs) in the population of hemodialysis patients. This is a retrospective study that included a large sample size. The main conclusion is that the use of PPIs is associated with a higher risk of all-cause mortality, especially in patients with a high burden of comorbidities, but the impact of PPI side effects in this group of patients cannot be ruled out. The main drawback is that this information is practically irrelevant from a practical point of view, as no in-depth analysis of PPI indications and causes of death has been performed. We do not know whether these drugs were used for, for example, gastritis found in gastroscopy or prophylactically due to, taking glucocorticosteroids. We also do not know whether the cause of death was infectious complications or cardiovascular disease.

Such a limitation was also emphasized by the authors, which unfortunately reduces the value of this work.

After reading the manuscript, the reader still does not know whether PPIs are friends or foes of HD patients who have a higher burden of gastrointestinal problems. Of course, I understand that with such a large sample it is difficult to collect such data, but without it it is impossible to draw any practical conclusions.

Answer: Thank you for your comment. We performed additional subgroup analyses based on the presence of coronary artery disease (CAD) or gastrointestinal diseases (GID), as well as steroid or antiplatelet agent usage. CAD was defined as the presence of a procedural code for percutaneous transluminal coronary angioplasty within 1 year prior to the HD Quality Assessment Program (M6551, M6552, M6561-4, M6571, and M6572). GID was defined using procedural code for the upper gastrointestinal tract, as the ICD-10 disease code was found to be inaccurate in GID diagnosis. GID was defined as the presence of a procedural code for endoscopic treatment of upper gastrointestinal perforation (Q7660), simple closure of perforated stomach and duodenum (Q2540), surgical clipping (Q2510), endoscopic hemostasis of upper gastrointestinal tract (Q7620), and embolization (M6644). Steroid usage was defined as an oral steroid prescription for 30 days during the assessment period, and antiplatelet agent usage was defined as a prescription for aspirin, clopidogrel, or ticlopidine for 30 days during the assessment period.

The results of the subgroup analysis based on the presence of CAD or GID, or steroid or antiplatelet agent usage are as follows:

Table S2. Cox regression analyses for patient survival using based on subgroups

Univariate

Multivariate

HR (95% CI)

P

HR (95% CI)

P

Patients with coronary artery disease

  Ref.: Group 1

    Group 2

0.93 (0.76–1.14)

0.506

0.91 (0.70–1.18)

0.477

Group 3

1.00 (0.84–1.20)

0.958

1.09 (0.87–1.38)

0.454

  Ref.: Group 2

    Group 3

1.08 (0.84–1.37)

0.552

1.20 (0.89–1.62)

0.232

Patients without coronary artery disease

  Ref.: Group 1

    Group 2

1.07 (1.02–1.12)

0.009

0.99 (0.94–1.05)

0.814

Group 3

1.27 (1.22–1.33)

<0.001

1.09 (1.04–1.15)

<0.001

  Ref.: Group 2

    Group 3

1.19 (1.13–1.27)

<0.001

1.10 (1.02–1.18)

0.010

Patients with steroid usage

  Ref.: Group 1

    Group 2

0.98 (0.79–1.21)

0.832

0.86 (0.66–1.11)

0.243

Group 3

1.27 (1.08–1.49)

0.005

1.05 (0.85–1.29)

0.645

  Ref.: Group 2

    Group 3

1.30 (1.02–1.65)

0.032

1.23 (0.92–1.64)

0.171

Patients without steroid usage

  Ref.: Group 1

    Group 2

1.08 (1.03–1.29)

0.003

1.00 (0.94–1.06)

0.952

Group 3

1.27 (1.22–1.32)

<0.001

1.09 (1.03–1.15)

<0.001

  Ref.: Group 2

    Group 3

1.18 (1.11–1.25)

<0.001

1.09 (1.02–1.17)

0.018

Patients with gastrointestinal disease

  Ref.: Group 1

    Group 2

0.97 (0.72–1.32)

0.862

0.78 (0.53–1.14)

0.195

Group 3

1.15 (0.91–1.44)

0.235

0.91 (0.68–1.21)

0.518

  Ref.: Group 2

    Group 3

1.18 (0.84–1.66)

0.346

1.17 (0.77–1.79)

0.457

Patients without gastrointestinal disease

  Ref.: Group 1

    Group 2

1.07 (1.03–1.13)

0.003

1.00 (0.94–1.06)

0.959

Group 3

1.27 (1.22–1.33)

<0.001

1.10 (1.04–1.15)

<0.001

  Ref.: Group 2

    Group 3

1.19 (1.12–1.26)

<0.001

1.10 (1.02–1.18)

0.010

Patients with antiplatelet agent usage

  Ref.: Group 1

    Group 2

1.01 (0.94–1.08)

0.840

0.96 (0.87–1.05)

0.346

Group 3

1.22 (1.14–1.30)

<0.001

1.06 (0.98–1.15)

0.164

  Ref.: Group 2

    Group 3

1.21 (1.10–1.32)

<0.001

1.11 (0.99–1.24)

0.080

Patients without antiplatelet agent usage

  Ref.: Group 1

    Group 2

1.10 (1.04–1.17)

0.002

1.02 (0.95–1.10)

0.539

Group 3

1.26 (1.20–1.33)

<0.001

1.12 (1.06–1.19)

<0.001

  Ref.: Group 2

    Group 3

1.14 (1.06–1.23)

<0.001

1.10 (1.00–1.20)

0.040

Multivariate analysis was adjusted for age, sex, underlying cause of end-stage renal disease, Charlson Comorbidity Index score, vascular access type, hemodialysis vintage, ultrafiltration volume, Kt/Vurea, hemoglobin, serum albumin, serum creatinine, serum phosphorus, serum calcium, systolic blood pressure, diastolic blood pressure, and use of anti-hypertensive drugs, aspirin, and statins, and was performed using enter mode.

Abbreviations: CI, confidence interval; ESRD, end-stage renal disease; HR, hazard ratio.

Multivariate analyses showed that PPI usage was not associated with patient survival in patients with CAD or GID, and in those who used steroids or antiplatelet agents. However, Group 3 showed poorer survival rates than that of other groups among patients without CAD or GID, and in those who did not use steroids or antiplatelet agents. These results reveal PPI usage is not always associated with high mortality in patients undergoing HD. We were unable to confirm whether the presence of CAD or GID, or steroid or antiplatelet agent usage are absolute indications of PPI usage. However, steroid or antiplatelet agent usage, or the presence of CAD or GID may partially explain the reason for PPI usage. The association between PPI usage and mortality was not observed in patients with a higher burden of gastrointestinal problems, including those with CAD, GID, or who were using steroid or antiplatelet agent. This suggests that PPI use may be acceptable only in these specific patient subgroups.

We have added these comments to the Methods, Results, Discussion, and Conclusion sections of our revised manuscript.

However, apart from this fundamental remark, I have one minor. I think the Table 2 needs to be corrected because the lines are shifted and the HR values seem to be misplaced.

Answer: Thank you for your comment. We have repositioned Table 2 according to the reviewer’s suggestion.

Reviewer 2 Report

To the Editor of the Journal of Clinical Medicine

It was with interest that I read Seok-Hui et al. manuscript “Effects of proton pump inhibitors on patient survival in patients undergoing maintenance hemodialysis”, a retrospective study aimed to evaluate the effect of PPIs on patient survival in a large sample of 54,903 patients on maintenance HD. Patients were divided in three groups according to the PPI prescription (no PPIs, < 90 days use and ≥ 90 days use, and 5-year survival rates was analyzed, The authors concluded that long-term PPI use was associated with high mortality.

COMMENTS:

PPIs are abundantly prescribed and are reported to be associated with a number of adverse health and kidney outcomes, including hypomagnesemia, acute kidney injury, acute interstitial nephritis, incident chronic kidney disease, kidney disease progression, kidney failure, and increased risk for all-cause mortality and mortality due to chronic kidney disease also in patients receiving maintenance hemodialysis.

The topic, not completely original, is in any case well treated and has its strong point in the sample size and in any case adds something useful for the reader on the correct prescription of PPIs, and the possible risks, not only related to the progression of renal insufficiency due to chronic tubulointerstitial nephritis, but also of increased mortality in patients on hemodialysis therapy. There is no doubt that in many countries there is an over-prescription of PPIs, perhaps ignoring the possible side effects also on renal toxicity so it is difficult to determine whether PPIs use may be prescribed surely in patients undergoing HD.

One observation that the authors should comment on is that, in patients with indications for PPIs, perhaps as gastric coverage in patients on antiplatelet therapy, or sometimes with dual antiplatelet therapy, PPIs therapy is prescribed due to the risk cardio-vascular or after previous coronary or peripheral vascular interventions, as described in Group 3.

In Table 3, baseline characteristics, the age was relatively young, when compared with other patients on hemodialysis in other occidental countries, and, surprisingly, no patients had central venous catheter as vascular access for dialysis, a strange if we consider that diabetes was the cause of ESKF in almost 50% in all 3 groups.

I would like to add these comments to the authors:

a- is it possible to know the survival in the subgroup of patients with coronary artery disease treated with antiplatelets and PPIs?

b- section 2. Materials and Methods, 2.1 Data source: The numbers of patients.. is it possible to insert a summary table?

c. Results: I observe that the Hb values are not at the target generally accepted by the guidelines (Hb=10.6-10.7). Any comments by the authors?

Author Response

PPIs are abundantly prescribed and are reported to be associated with a number of adverse health and kidney outcomes, including hypomagnesemia, acute kidney injury, acute interstitial nephritis, incident chronic kidney disease, kidney disease progression, kidney failure, and increased risk for all-cause mortality and mortality due to chronic kidney disease also in patients receiving maintenance hemodialysis.

 The topic, not completely original, is in any case well treated and has its strong point in the sample size and in any case adds something useful for the reader on the correct prescription of PPIs, and the possible risks, not only related to the progression of renal insufficiency due to chronic tubulointerstitial nephritis, but also of increased mortality in patients on hemodialysis therapy. There is no doubt that in many countries there is an over-prescription of PPIs, perhaps ignoring the possible side effects also on renal toxicity so it is difficult to determine whether PPIs use may be prescribed surely in patients undergoing HD.

One observation that the authors should comment on is that, in patients with indications for PPIs, perhaps as gastric coverage in patients on antiplatelet therapy, or sometimes with dual antiplatelet therapy, PPIs therapy is prescribed due to the risk cardio-vascular or after previous coronary or peripheral vascular interventions, as described in Group 3.

Answer: Thank you for your comments. We performed additional subgroup analyses based on the presence of coronary artery disease (CAD) or gastrointestinal diseases (GID), as well as steroid or antiplatelet agent usage. CAD was defined as the presence of a procedural code for percutaneous transluminal coronary angioplasty within 1 year prior to the HD Quality Assessment Program (M6551, M6552, M6561-4, M6571, and M6572). GID was defined using procedural code for the upper gastrointestinal tract, as the ICD-10 code was found to be inaccurate in GID diagnosis. GID was defined as the presence of a procedural code for endoscopic treatment of upper gastrointestinal perforation (Q7660), simple closure of perforated stomach and duodenum (Q2540), surgical clipping (Q2510), endoscopic hemostasis of upper gastrointestinal tract (Q7620), and embolization (M6644). Steroid usage was defined as an oral steroid prescription for 30 days during the assessment period, and antiplatelet agent usage was defined as a prescription for aspirin, clopidogrel, or ticlopidine for 30 days during the assessment period.

The results of the subgroup analysis based on the presence of CAD or GID, or steroid or antiplatelet agent usage are as follows:

Table S2. Cox regression analyses for patient survival based on subgroups

Univariate

Multivariate

HR (95% CI)

P

HR (95% CI)

P

Patients with coronary artery disease

  Ref.: Group 1

    Group 2

0.93 (0.76–1.14)

0.506

0.91 (0.70–1.18)

0.477

Group 3

1.00 (0.84–1.20)

0.958

1.09 (0.87–1.38)

0.454

  Ref.: Group 2

    Group 3

1.08 (0.84–1.37)

0.552

1.20 (0.89–1.62)

0.232

Patients without coronary artery disease

  Ref.: Group 1

    Group 2

1.07 (1.02–1.12)

0.009

0.99 (0.94–1.05)

0.814

Group 3

1.27 (1.22–1.33)

<0.001

1.09 (1.04–1.15)

<0.001

  Ref.: Group 2

    Group 3

1.19 (1.13–1.27)

<0.001

1.10 (1.02–1.18)

0.010

Patients with steroid usage

  Ref.: Group 1

    Group 2

0.98 (0.79–1.21)

0.832

0.86 (0.66–1.11)

0.243

Group 3

1.27 (1.08–1.49)

0.005

1.05 (0.85–1.29)

0.645

  Ref.: Group 2

    Group 3

1.30 (1.02–1.65)

0.032

1.23 (0.92–1.64)

0.171

Patients without steroid usage

  Ref.: Group 1

    Group 2

1.08 (1.03–1.29)

0.003

1.00 (0.94–1.06)

0.952

Group 3

1.27 (1.22–1.32)

<0.001

1.09 (1.03–1.15)

<0.001

  Ref.: Group 2

    Group 3

1.18 (1.11–1.25)

<0.001

1.09 (1.02–1.17)

0.018

Patients with gastrointestinal disease

  Ref.: Group 1

    Group 2

0.97 (0.72–1.32)

0.862

0.78 (0.53–1.14)

0.195

Group 3

1.15 (0.91–1.44)

0.235

0.91 (0.68–1.21)

0.518

  Ref.: Group 2

    Group 3

1.18 (0.84–1.66)

0.346

1.17 (0.77–1.79)

0.457

Patients without gastrointestinal disease

  Ref.: Group 1

    Group 2

1.07 (1.03–1.13)

0.003

1.00 (0.94–1.06)

0.959

Group 3

1.27 (1.22–1.33)

<0.001

1.10 (1.04–1.15)

<0.001

  Ref.: Group 2

    Group 3

1.19 (1.12–1.26)

<0.001

1.10 (1.02–1.18)

0.010

Patients with antiplatelet agent usage

  Ref.: Group 1

    Group 2

1.01 (0.94–1.08)

0.840

0.96 (0.87–1.05)

0.346

Group 3

1.22 (1.14–1.30)

<0.001

1.06 (0.98–1.15)

0.164

  Ref.: Group 2

    Group 3

1.21 (1.10–1.32)

<0.001

1.11 (0.99–1.24)

0.080

Patients without antiplatelet agent usage

  Ref.: Group 1

    Group 2

1.10 (1.04–1.17)

0.002

1.02 (0.95–1.10)

0.539

Group 3

1.26 (1.20–1.33)

<0.001

1.12 (1.06–1.19)

<0.001

  Ref.: Group 2

    Group 3

1.14 (1.06–1.23)

<0.001

1.10 (1.00–1.20)

0.040

Multivariate analysis was adjusted for age, sex, underlying cause of end-stage renal disease, Charlson Comorbidity Index score, vascular access type, hemodialysis vintage, ultrafiltration volume, Kt/Vurea, hemoglobin, serum albumin, serum creatinine, serum phosphorus, serum calcium, systolic blood pressure, diastolic blood pressure, and use of anti-hypertensive drugs, aspirin, and statins, and was performed using enter mode.

Abbreviations: CI, confidence interval; ESRD, end-stage renal disease; HR, hazard ratio.

Multivariate analyses showed that PPI usage was not associated with patient survival in patients with CAD or GID, and in those who used steroids or antiplatelet agents. However, Group 3 showed poorer survival rates than that of other groups among patients without CAD or GID, and in those who did not use steroids or antiplatelet agents. These results reveal PPI usage is not always associated with high mortality in patients undergoing HD. We were unable to confirm whether the presence of CAD or GID, or steroid or antiplatelet agent usage are absolute indications of PPI usage. However, steroid or antiplatelet agent usage, or the presence of CAD or GID may partially explain the reason for PPI usage. The association between PPI usage and mortality was not observed in patients with a higher burden of gastrointestinal problems, including those with CAD, GID, or who were using steroid or antiplatelet agent. This suggests that PPI use may be acceptable only in these specific patient subgroups.

We have added these comments to the Methods, Results, Discussion, and Conclusion sections of our revised manuscript.

In Table 3, baseline characteristics, the age was relatively young, when compared with other patients on hemodialysis in other occidental countries, and, surprisingly, no patients had central venous catheter as vascular access for dialysis, a strange if we consider that diabetes was the cause of ESKF in almost 50% in all 3 groups.

 Answer: Thank you for your comment. Patients with an HD catheter may have reversible renal diseases and/or other acute complications. Furthermore, HD catheters themselves are also associated with various complications, which can be confounding factors for accurately determining all-cause mortality. Therefore, we have excluded all patients with HD catheters in our cohort; all the remaining patients in our study underwent HD via an AVF or AVG.

In our study, the mean ages of Groups 1, 2, and 3 were 59.8, 60.2, and 62.6 years old, respectively. These ages are approximately 2–3 years younger than those from Japanese data and other Korean registries [1,2]. The National HD Quality Assessment Program include the patients who underwent HD in the outpatient department, excluding those admitted for acute events that could influence HD quality. Elderly patients undergoing HD have a high prevalence of physical disability. Some of these elderly patients underwent HD on admission to convalescent hospitals; such patients would be expected to have a higher prevalence of admission due to acute/chronic events during each assessment compared to younger patients. Thus, the exclusion of admitted patients may have contributed to the relatively younger age observed in our cohort.

In addition, DM was the most common cause of end-stage renal disease in our cohort. In the Korean registry, the frequency of DM as the cause of end-stage renal disease was 19.5% in 1992, 30.5% in 2005, and 47.0% in 2021, respectively [1]. The prevalence of DM is steeply increasing. Considering that 9–11% of our cohort have an unknown cause for end-stage renal disease, approximately half of HD patients are undergoing HD due to DM.

We have added these comments to the Methods and Discussion sections of the revised manuscript.

References

[1] Nitta K, Goto S, Masakane I, Hanafusa N, Taniguchi M, Hasegawa T, et al. Annual dialysis data report for 2018, JSDT Renal Data Registry: survey methods, facility data, incidence, prevalence, and mortality. Ren Replace Ther. 2020;6(1):41.

[2] ESRD Registry Committee: Korean Society of Nephrology. Current Renal Replacement Therapy in Korea, 2022. Available at: https://ksn.or.kr/bbs/index.php?code=report. Assessed May 1, 2023.

I would like to add these comments to the authors:

a- is it possible to know the survival in the subgroup of patients with coronary artery disease treated with antiplatelets and PPIs?

Answer: Thank you for your comment. We have performed subgroup analyses based on the presence of CAD or antiplatelet agent usage. Detailed explanations have been presented in our answers to the reviewers’ previous comments.

b- section 2. Materials and Methods, 2.1 Data source: The numbers of patients.. is it possible to insert a summary table?

Answer: Thank you for your comment. We have added a flowchart figure showing the study design, as follows:

Figure S1. Study flow chart.

  1. Results: I observe that the Hb values are not at the target generally accepted by the guidelines (Hb=10.6-10.7). Any comments by the authors?

Answer: Thank you for your comment. In our study, mean hemoglobin levels ranged from 10.5–10.6 g/dL. Previous guidelines have recommended target hemoglobin levels of 10–11.5 g/dL, with levels above 13 g/dL not being recommended [1]. In our cohort, the number of patients with hemoglobin levels < 10 g/dL was 7585 (13.8%). The relatively high proportion of patients with hemoglobin levels below the target range may be due to difficulty in obtaining erythropoietin-stimulating agents. In South Korea, insurance covers erythropoiesis-stimulating agents in patients undergoing HD with hemoglobin levels 11 g/dL. If the patient undergoing HD has hemoglobin levels > 11 g/dL, the clinician may discontinue the erythropoiesis-stimulating agent, even though the hemoglobin levels are expected to decrease. Therefore, maintaining a stable hemoglobin level > 10~11 g/dL is challenging due to insurance limitations. Nevertheless, the mean hemoglobin levels in all three groups of our cohort remained within the target range.

We have added these comments to the Discussion section of the revised manuscript.

Reference

[1] Drüeke, T.B.; Parfrey, P.S. Summary of the KDIGO guideline on anemia and comment: reading between the guidelines. Kidney Int. 2012 Nov;82(9):952-60.
